# Whole genome sequence analysis of pulmonary function and COPD in 19,996 multi-ethnic participants

Xutong Zhao ⓘ et al.[#]

Chronic obstructive pulmonary disease (COPD), diagnosed by reduced lung function, is a leading cause of morbidity and mortality. We performed whole genome sequence (WGS) analysis of lung function and COPD in a multi-ethnic sample of 11,497 participants from population- and family-based studies, and 8499 individuals from COPD-enriched studies in the NHLBI Trans-Omics for Precision Medicine (TOPMed) Program. We identify at genome-wide significance 10 known GWAS loci and 22 distinct, previously unreported loci, including two common variant signals from stratified analysis of African Americans. Four novel common variants within the regions of *PIAS1*, *RGN* (two variants) and *FTO* show evidence of replication in the UK Biobank (European ancestry $n \sim 320,000$), while colocalization analyses leveraging multi-omic data from GTEx and TOPMed identify potential molecular mechanisms underlying four of the 22 novel loci. Our study demonstrates the value of performing WGS analyses and multi-omic follow-up in cohorts of diverse ancestry.

---

[#]A list of authors and their affiliations appears at the end of the paper.

Lung function is an important measure of health and an independent predictor of morbidity and mortality in the general population[1,2]. Chronic obstructive pulmonary disease (COPD) is characterized by chronic airflow limitation typically in response to noxious environmental stimuli, is the fourth leading cause of death in the United States[3,4], the third leading cause of death world-wide[5], and has shown continued increases in prevalence in recent years[6]. COPD is diagnosed by spirometric decreases in lung function, namely forced expiratory volume in one second ($FEV_1$) and its ratio to forced vital capacity ($FEV_1$/FVC). While the main risk factor for COPD is cigarette smoking, the risk of COPD also increases with age, and can progress even after smoking cessation[7]. Despite the enormous burden of COPD, there are currently no pharmacologic therapies that convincingly slow progression of disease or reduce mortality, and there is, therefore, an unmet need for new therapeutics. Since the genetic risk factors for COPD are poorly understood, discovery of disease-associated loci can elucidate pathogenetic mechanisms and identify putative molecular targets.

COPD has substantial heritability, even after accounting for differences in cigarette smoking behavior, with estimates ranging from 35–60%[8–10]. Quantitative measures of lung function are also similarly heritable in the general population with over 40% of variation in $FEV_1$, FVC and $FEV_1$/FVC attributable to genetic factors[11]. Genome-wide association studies (GWAS) have identified numerous loci for both COPD and pulmonary function. Studies led by the Cohorts for Heart and Aging Research in Genomic Epidemiology (CHARGE)/SpiroMeta[12–14] and the UK Biobank[15], focused primarily on European ancestry subjects, identified over a hundred genetic loci that contain common single nucleotide polymorphisms (SNPs) significantly associated with $FEV_1$, FVC, $FEV_1$/FVC. A GWAS of pulmonary function from the CHARGE consortium, combining participants of European ancestry, African ancestry and Hispanic/Latino participants identified an additional 50 loci in ancestry specific and multi-ethnic analyses[16]. More recently, a large scale GWAS including >400,000 European ancestry participants from UK Biobank and the SpiroMeta consortium identified an additional 139 signals for pulmonary function traits[17]. Thus, increased ethnic diversity as well as order of magnitude increases in sample size have proven to be effective ways of identifying novel genetic loci for pulmonary function.

The largest published GWAS of COPD to date, from the International COPD Genetics Consortium, included 35,735 cases and 222,076 controls from the UK Biobank and participants from over twenty other studies in the International COPD Genetics Consortium, and identified 82 genome-wide significant loci for COPD[18]. Of the 35 novel COPD-associated loci identified in that study, 13 were associated with lung function in independent samples from the SpiroMeta consortium after Bonferroni correction for multiple testing, and an additional 14 showed nominal association with lung function ($P < 0.05$). Thus, despite a high degree of overlap between genetic loci for COPD and lung function, there may be advantages to studying quantitative traits versus dichotomous outcomes, and both approaches can be fruitful.

Previous GWAS have been limited in part by the sample sizes and race/ethnic representation of available reference panels (e.g. HapMap[19] or 1000 Genomes[20]), resulting in missing information on both common and rare variants, particularly in African Americans and Hispanics. Rare variants affect COPD susceptibility. For example, severe alpha-1 antitrypsin deficiency has been recognized for decades as a genetic cause of COPD[10]. However, additional rare variants have been difficult to identify.

To address these limitations, we leveraged deep sequencing in the NHLBI Trans-Omics for Precision Medicine (TOPMed) Program to perform the first large-scale, multi-ethnic whole genome sequence (WGS) analysis of pulmonary function and COPD. We report at genome-wide significance 10 known GWAS loci and 22 distinct, previously unreported loci. We demonstrate evidence of replication with consistent direction of effect for four common variants, supportive evidence through colocalization for two additional signals, and a rare variant of large effect on reduced lung function ($FEV_1$/FVC ratio). In gene-based analysis, we report association with increased $FEV_1$/FVC in individuals with rare variants in the gene *ARHGEF17*. Several of the novel loci that we identify in our study were neither included on GWAS chips nor well imputed by existing sources, highlighting the importance of our WGS approach.

## Results

**Participant characteristics**. Our study sample included 19,996 participants, with 11,497 participants from population- and family-based studies, as well as 8499 participants from COPD-enriched studies (Table 1, Fig. 1). Using participant self-reported race/ethnicity, 12,316 and 6450 participants were categorized as non-Hispanic White or African American, respectively. The remaining 1224 participants represented Hispanic, Asian and other races/ethnicities. The combined samples included 4466 moderate-to-severe COPD cases and 1739 severe COPD cases. Among these, 1279 moderate-to-severe and 220 severe COPD cases were contributed by population- and family-based cohorts, and the remaining COPD cases were from the COPD-enriched studies (Supplementary Data 1).

**Table 1 Summary of the 19,996 study-participants included in analyses.**

| Stratum | Study | Sample size (COPD cases) | | | |
|---|---|---|---|---|---|
| | | Non Hispanic White | African American | Other* | All combined |
| Population-and family-based | Atherosclerosis risk in communities (ARIC) | 3075 (554) | 181 (16) | — | 3256 (570) |
| | Cleveland Family Study (CFS) | 373 (24) | 346 (34) | — | 719 (58) |
| | Cardiovascular Health Study (CHS) | 39 (8) | — | 8 (2) | 47 (10) |
| | Framingham Heart Study (FHS) | 1835 (187) | — | — | 1835 (187) |
| | Jackson Heart Study (JHS) | — | 2,388 (121) | — | 2388 (121) |
| | Multi-Ethnic Study of Atherosclerosis (MESA) | 1224 (173) | 804 (84) | 1224 (76) | 3252 (333) |
| | Total | 6546 (946) | 3719 (255) | 1232 (78) | 11,497 (1279) |
| COPD-enriched | Genetic Epidemiology of COPD (COPDGene) | 5713 (2416) | 2731 (717) | — | 8444 (3133) |
| | Boston Early Onset COPD (EOCOPD) | 55 (54) | — | — | 55 (54) |
| | Total | 5768 (2470) | 2731 (717) | — | 8499 (3187) |
| Combined | Total | 12,314 (3416) | 6450 (972) | 1232 (78) | 19,996 (4466) |

*The total number of CHS participants includes 8 African American individuals who were not included in stratified analysis of African Americans only due to the small number.

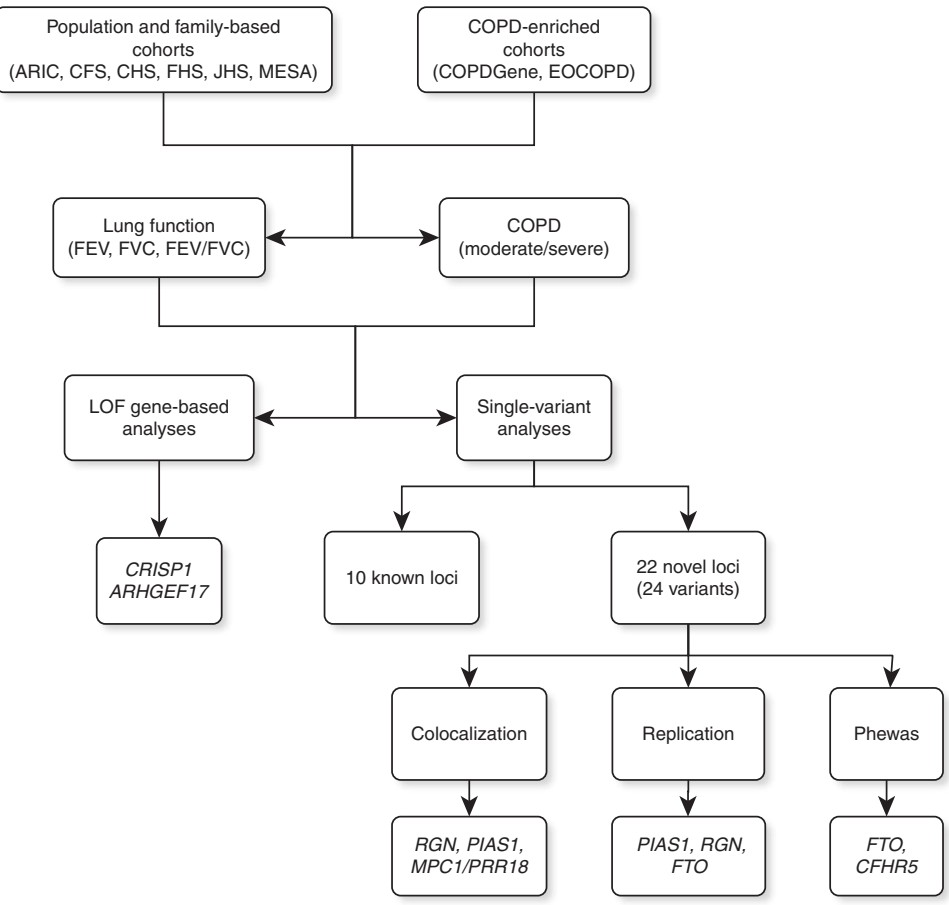

**Fig. 1 Overview of workflow for the study.** Whole genome sequence analysis of lung function and COPD was carried out in TOPMed participants from population- and family-based studies, as well as in COPD-enriched studies. We performed gene-based analysis of pLOF variants as well as single variant analysis. Genetic variants and loci identified by single variant analysis were further examined for colocalization with gene expression (eQTL) and methylation (mQTL) traits, as well as through replication and phenome-wide association studies (Phewas). Note: Novel loci demonstrating evidence of colocalization with eQTL are labeled according to the corresponding gene expression targets. All other loci are labeled using the nearest gene as indicated in Tables 2 and 3.

**Single variants overlapping known GWAS loci.** In single variant analysis, we observed at genome-wide significance the signals of association for 10 known pulmonary function and COPD GWAS loci. In particular, we confirmed association within the region of *HTR4* in population- and family-based analysis, associations within the regions of *CHRNA3/5*; *FAM13A*; *EEFSEC*; *RIN3*; and *HHIP* in analysis of COPD-enriched samples, and associations in/ near *CHRNA3/5*; *GSTCD*; *AGER*; *DSP*; *RIN3*; *HHIP*; *FAM13A*; *THSD4*; and *EEFSEC* in combined analysis incorporating both population-based and COPD-enriched strata (Supplementary Data 2). While we used a cutoff of LD R-squared >0.2 to group genome-wide significant variants identified in our study with those from prior literature, we note that all of the variants identified as overlapping with known GWAS signals demonstrated a high level of LD, having R-squared >0.7 with at least one previously reported variant.

For variants identified in previous GWAS studies of lung function[15–17,21] and COPD[18,22,23], we further examined the evidence of association in our study overall, as well as in focused stratified analyses. Our combined analysis across all race/ethnic-groups identified 32, 30, 75 and 55 nominally significant signals (at $P < 0.05$) out of 104, 122, 181 and 156 variants examined in association with FEV1, FVC, FEV1/FVC ratio, and COPD, respectively (Supplementary Data 3). The numbers of statistically significant signals were reduced in stratified analyses, reflecting in

part the lower power in these smaller sample sizes (Supplementary Fig. 1). The directions of effect that we observed were largely consistent with those reported in previous studies, both in combined analysis as well as in race/ethnic-statified analysis (Supplementary Data 3–8).

**Twenty-four novel variants spanning 22 distinct loci.** Among our genome-wide significant single variant findings, we report 27 association signals across all strata included in our analyses, covering 24 distinct variants (Supplementary Data 2, Supplementary Figs. 2, 3). After grouping together variants within 5 Mb of each other and with LD R-squared >0.2 (examined overall and separately within our TOPMed White or African American samples), there were 22 distinct loci among our novel findings. These loci include two common autosomal variant signals identified in stratified analysis of African Americans (in/near *GALNT18* and *CMIP*), five common autosomal variants identified in Whites and/or in pooled analyses across all race/ethnicities, three distinct common variant signals on the X chromosome, and nine rare variants with minor allele frequencies less than 1%. Sex-stratified analysis of the variants on the X chromosome showed they had consistent directions of effect in males and females (Supplementary Data 9), with two of the variants showing notably larger effects in males compared to females (rs142755000: heterogeneity I-squared = 81.3, and

rs5953026: heterogeneity I-squared = 91.4). Among the nine novel rare variants, five were intronic and four were intergenic (Supplementary Data 10). The novel rare $FEV_1/FVC$-associated variant rs184101688 (MAF = 0.001, beta = $-9.42$, $P = 1.3 \times 10^{-8}$ for SAIGE score test) is located ~76 kb from a previously identified common variant (rs4318980, MAF = 0.42 in 1000 Genomes EUR) in the region of $C1GALT1$[17].

We identified one novel association with $FEV_1/FVC$ at rs7188378, located within the third intron of $FTO$ (T > C, combined all race/ethnicities MAF = 0.489, beta = $-0.59$ and $P = 4.9 \times 10^{-8}$ for SAIGE score test), located 29 kb away from rs35420030 (MAF = 0.06 in 1000 Genomes EUR) recently reported[17] in association with FEV1/FVC ratio. Conditional analysis demonstrated the association with rs7188378 was largely independent of rs35420030 ($P = 2.3 \times 10^{-7}$ for conditional association test). Our TOPMed lead variant is located about 86 kb from the $FTO$ region variant rs9939609 (MAF = 0.42 in 1000 Genomes EUR) reported in the first publication that linked the $FTO$ locus to body mass index [BMI][24]. Although the physical distance separating the lead $FTO$ variant identified in our TOPMed WGS from the previously reported $FTO$ variant is large, the allele frequencies are similar and the BMI locus is known to lie in a region of long-range LD[25]. Therefore, we examined LD between our lead variant rs7188378 and the previously reported BMI variant rs9939609 and found LD R-squared <0.02 in our TOPMed White sample as well as in all race/ethnicities. We further performed a sensitivity analysis examining genetic association with additional covariate adjustment for weight, and noted only a modest attenuation in the association signal under this model (T > C, beta = $-0.54$ and $P = 3.70 \times 10^{-7}$ for SAIGE score test).

**Single variant effect estimates across subgroups**. Our newly identified variants demonstrated largely consistent directions of effect across cohorts contributing to the reported results (Supplementary Data 2). With the exception of the result for rs35917906 in which JHS showed direction opposite that seen in all other cohorts (Supplementary Fig. 3p), all other inconsistencies in direction of effect among cohorts included in specific strata for the reported discovery analyses came from CFS, CHS and EOCOPD which had substantially smaller sample sizes compared to the other included studies (Table 1).

For those variants identified in combined analysis including both population- and family-based cohorts and COPD-enriched studies, we observed several instances in which the effect sizes were substantially larger in the COPD-enriched studies compared to population-based cohorts. For example, rs7188378, which was associated with $FEV_1/FVC$ in combined analysis, had larger effect sizes in COPDGene study than most of the other studies (Supplementary Fig. 3y; COPDGene: beta = $-1.07$ vs. ARIC: beta = $-0.26$; FHS: beta = $-0.14$; JHS: beta = $-0.39$; MESA: beta = $-0.22$). We do not compare directly with EOCOPD, CFS and CHS here due to their small sample sizes.

For novel signals identified in race/ethnic stratified analyses of particular subgroups, comparing the observed directions of effect in the discovery group to other subgroups in TOPMed further suggested potential heterogeneity across groups. For example, for the variant rs74469188 identified in analysis of COPD-enriched African Americans, while the effect allele C was positively associated with FVC in COPDGene African Americans, the only other subgroups demonstrating positive effect estimates among the other TOPMed cohorts were both African American (from MESA and ARIC; see Supplementary Fig. 3h).

**Genomic inflation factors**. Population structure was well controlled in both stratified and combined single variant WGS

analyses, with lambda values less than 1.04 for all quantitative trait analyses (Supplementary Table 1). For dichotomous trait analyses of moderate-to-severe and severe COPD, lambda values were all less than 1.06 in stratified analysis of Whites or African Americans, and less than 1.08 in the combined analysis (Supplementary Table 1).

**Additional signals identified by conditional analysis**. Conditioning on the lead signals for novel variants (Supplementary Data 2), we identified a secondary variant rs142712254 associated with FVC in the combined analysis of all race/ethnicities (conditional beta = $-0.10$ and $P = 1.91 \times 10^{-6}$ for conditional association test) after conditioning on the lead variant rs35917906 near $RGN$ on chromosome X. This lead variant at this secondary signal, rs142712254, is in LD with the variant rs182915372 that we identified in population- and family-based analysis of FVC (LD R-squared = 0.68 in our TOPMed samples using all race/ethnicities). In conditional analysis accounting for variants within known loci (Supplementary Data 2), we also found a secondary variant, rs9920270, around the $CHRNA3/5$ region variant rs12914385 associated with severe COPD in COPD-enriched analysis (conditional beta = 0.28 and $P = 2.25 \times 10^{-7}$ for conditional association test; Supplementary Data 11). Among our findings at a more liberal threshold, we identified a suggestive association with rs34712979 associated with $FEV_1$ in the combined analysis at the $GSTCD$ locus (conditional beta = $-0.03$ and $P = 8.11 \times 10^{-6}$ for conditional association test), confirming a previous report[26].

**Whole genome sequence versus imputed genotypes**. For the novel variants (Supplementary Data 2), we observed generally very high R-squared between TOPMed Freeze 5b WGS variant calls and those obtained by imputation of genome-wide genotypes for common variants, suggesting that common variants with MAF greater than ~3–5% were examined effectively in prior GWAS efforts leveraging imputation to existing reference panels. However, for infrequent and rare variants, notably those with MAF less than 1%, many variants did not have imputed genotypes available in imputation based on the 1000 Genomes (Phases 1 and 3) or HRC, and those imputed genotypes that were available generally had R-squared of ~30–80% with WGS variant calls from TOPMed (Supplementary Data 12).

**Annotation of the identified variants**. Among our 24 distinct novel variants (Supplementary Data 2), we identified 12 variants located within intronic gene regions suggesting they may play roles in regulation of the overlapping genes. We further identified, three non-coding exon variants, rs4076943, rs74469188, rs7046490, lying within the regions of the pseudogene $MTND5P21$, the mature microRNA $MIR6504$, and the long non-coding RNA $RP11$-$130C19.3$, respectively (Supplementary Data 10).

**Rare putative loss of function (pLOF) variant burden analysis**. A burden of rare pLOF variants in $CRISP1$ showed genome-wide significant association with reduced FVC in combined analysis of African Americans (SAIGE-GENE burden test $P = 1.8 \times 10^{-6}$; cumulative allele frequency $2.02 \times 10^{-3}$). Among the candidate genes near GWAS loci, a burden of rare pLOF variants in $ARHGEF17$ was significantly associated with increased $FEV_1/FVC$ in combined analysis (SAIGE-GENE burden test $P = 1.9 \times 10^{-4}$; cumulative allele frequency $3.8 \times 10^{-4}$). In follow-up of this result, we found the burden of $ARHGEF17$ pLOF variants showed nominal association with $FEV_1$ (beta = 0.015, SAIGE-GENE burden test $P = 0.012$), but not FVC (beta = 0.006, SAIGE-GENE

burden test $P = 0.42$) Details of the variants included in the gene burden are shown in Supplementary Data 13.

**Replication of four novel variants.** We examined association with PFT traits for novel variants identified by our discovery WGS analysis (24 variants, Supplementary Data 2) and conditional analyses (2 variants, Supplementary Data 11) in the UK Biobank (European ancestry $n = 321,047$; African ancestry $n = 4350$). and the Hispanic Community Health Study/Study of Latinos (HCHS/SOL; $n = 11,822$).

After Bonferroni correction for the number of variants under consideration for each trait, we identified statistically significant evidence of replication with direction of effect consistent with that seen in TOPMed for four novel variants representing three distinct signals in the region of *PIAS1*, *FTO* and *RGN* (Table 2, Supplementary Data 14). The variant rs74469188 identified in COPD-enriched African Americans from TOPMed also showed statistically significant association with FVC in UK Biobank European ancestry samples (chr16: 81,611,365, *CMIP* intronic; UK Biobank BOLT-LMM $P = 1.4 \times 10^{-5}$), but the direction of effect was not consistent with that observed in TOPMed (Table 3). Among the four replicated variants, we also sought to determine whether any of them were associated with smoking behavior in the UK Biobank. The rs17308514 allele G, associated with decreased FVC, was also associated with decreased smoking initiation (BOLT-LMM $P = 7.2 \times 10^{-3}$). The rs35917906 allele T, associated with increased FVC, also demonstrated a nominal association with increased smoking cessation (BOLT-LMM $P = 0.043$; Supplementary Data 15).

In the analysis of the UK Biobank African ancestry samples, there were no statistically significant replication signals for pulmonary function (Supplementary Data 16), nor were there notable associations with smoking intensity traits after accounting for multiple comparisons (Supplementary Data 17). Similarly, we did not identify statistically significant associations of the novel TOPMed variants in relation to lung function in Hispanic/Latino participants from the HCHS/SOL[27] (Supplementary Data 18). We note that the African ancestry and admixed Hispanic cohorts were of relatively small sample size (Supplementary Fig. 4).

**Additional phenotypic consequences of the novel variants.** Phenome-wide analyses examining the novel variants from the TOPMed WGS analyses (Supplementary Data 2) for associations in the UK Biobank showed the variant rs7188378 intronic to *FTO* was associated with unspecified diffuse connective tissue disease (Table 2, Supplementary Data 19; UK Biobank MAF = 0.50; MAC in cases vs. controls = 3003 vs. 399,121; $P = 7.1 \times 10^{-14}$ for SAIGE score test) and diffuse diseases of connective tissue (MAC in cases vs. controls = 3771 vs. 399,096, $P = 6.4 \times 10^{-13}$ for SAIGE score test). The variant rs371740347 intronic to *CFHR5* was associated with "respiratory failure, insufficiency, arrest" (UK Biobank MAF = 0.01; MAC in cases vs. controls = 81 vs. 6,319; $P = 6.9 \times 10^{-5}$), though the allele increasing FVC was associated with increased risk of this phenotype (Table 3).

**Colocalization analysis suggests molecular mechanisms.** Through Bayesian colocalization analysis[28] using eQTLs from GTEx v7 across 48 tissues, we identified colocalization of multiple FVC-associated signals spanning a ~230 kb region on Xp11.3 (Tables 2–3, Supplementary Data 20) corresponding to the lead variants rs12556310, rs5953026 and rs35917906. In particular, rs12556310 and rs35917906 were colocalized with expression of *RGN*, *RNU6-1189P*, *USP11* and *NDUFB11* across multiple tissues, among which the colocalization of the signal at rs12556310 with *RGN* expression was observed in 30 different tissues, including lung (posterior probability for shared causal variant [PP4] = 0.924, Supplementary Fig. 5). The signal at rs5953026 was colocalized with *RGN* in testis only.

The FEV$_1$-associated locus at rs9295345 was colocalized with *MPC1* expression in three different tissues and with *PRR18* expression in tibial nerve. The same signal at rs9295345 was also colocalized with methylation of four correlated sites within the region in MESA whole blood, including cg06249499 for which the colocalization signal was consistent using whole blood from both Exams 1 and 5 in MESA (Supplementary Data 21; Supplementary Fig. 6). Measured levels of cg06249499 at baseline also demonstrated association with FEV$_1$ in MESA (multi-ethnic $P = 0.008$; Supplementary Table 2).

The FVC-associated signal at rs17308514 near the gene *PIAS1* demonstrated colocalization with one methylation site within the

---

**Table 2 Four distinct novel variants at three distinct signals\* with replication evidence.**

| TOPMed Discovery variant: rsid Chr:Pos (effect/other allele) | EAF (TOPMed stratum, race/ethnicity) | Trait (direction) | HC Beta (SE) P-value | Annotation | UK Biobank European ancestry replication\*\*: EffHC Beta (SE) P-value | Additional supporting evidence |
|---|---|---|---|---|---|---|
| rs17308514 15:68020833 (G/A) | 0.384 (Combined, All) | FVC (decreased) | HC = 8092 −0.04 (0.01) $P = 3.9 \times 10^{-8}$ | 5′ of *PIAS1* | EffHC = 160039 −0.007 (0.002) $P = 4.7 \times 10^{-3}$ | Colocalized methylation sites: cg00154119, cg20631419 |
| rs7188378 16:53872940 (C/T) | 0.475 (Combined, White) | FEV1/FVC (decreased) | HC = 6026 −0.79 (0.14) $P = 3.3 \times 10^{-8}$ | *FTO* intronic | EffHC = 158908 −0.009 (0.002) $P = 2.0 \times 10^{-4}$ | Previous GWAS variant\*\*\* rsid, Chr: Pos (effect/other allele): rs35420030 16:53901495 (C/T) EAF (discovery population): 0.052 (UK Biobank European ancestry) Trait (direction): FEV1/FVC (increased) Phenome-wide association\*\*\*\* trait (direction): unspecified diffuse connective tissue disease (increased) P-value = 7.1 × 10⁻¹⁴ |
| rs12556310 X:47087005 (G/C) | 0.440 (COPD-enriched, All) | FVC (increased) | HC = 1901 0.05 (0.01) $P = 3.3 \times 10^{-8}$ | *RGN* intronic | EffHC = 136386 0.006 (0.002) $P = 2.5 \times 10^{-3}$ | Colocalized gene expression traits: *RGN, RNU6-1189P, USP11, NDUFB11* |
| rs35917906 X:47100766 (T/C) | 0.489 (Combined, All) | FVC (increased) | HC = 3818 0.03 (0.01) $P = 1.5 \times 10^{-8}$ | 3′ of *RGN* | EffHC = 119997 0.009 (0.002) $P = 1.3 \times 10^{-5}$ | Colocalized gene expression traits: *RGN, RNU6-1189P, USP11, NDUFB11* |

Variants are reported based on genome-wide significance threshold of $P = 5 \times 10^{-8}$. All variant positions are presented based on Human Genome Build 38; EAF = effect allele frequency; HC = heterozygosity count; EffHC = effective heterozygosity count Genetic variant effects (betas) in TOPMed are reported for phenotypes under the heterogeneous variance model[56] such that the effect estimates reflect the scale of variance for FEV$_1$ (in L), FVC (in L) and FEV$_1$/FVC ratio (in %). P-values for genetic association in TOPMed as reported based on the SAIGE score test[55].
\*In determining the number of distinct signals, we grouped together two variants (rs12556310 and rs35917906) on chromosome X that were in LD with R-squared = 0.71 and 0.44 based on White and all race/ethnicities in our TOPMed sample, respectively.
\*\*Genetic variant effects for replication in UK Biobank are reported for inverse normal transformed residualized lung function traits, following the model used in Shrine et al[17]. P-values are reported based on the BOLT-LMM genetic association test[66].
\*\*\*Information on prior GWAS association reported based on result from Shrine et al.[17].
\*\*\*\*The phenome-wide association P-value is reported based on the SAIGE score test[55].

**Table 3 Additional genome-wide significant results with suggestive supporting evidence.**

| TOPMed variant rsid Chr:Pos (effect/other allele) | EAF (TOPMed stratum, race/ethnicity) | Trait (direction) | HC Beta (SE) P-value | Annotation | Supporting evidence type | Supporting evidence details |
|---|---|---|---|---|---|---|
| rs9295345 6:166400303 (T/G) | 0.725 (Combined, White) | FEV$_1$ (decreased) | HC = 4970 −0.08 (0.01) $P = 3.2 \times 10^{-8}$ | 3′ of *RPS6KA2*; 5′ of *MPC1* | *Colocalization* | Colocalized gene expression traits: *MPC1, PRR18* Colocalized methylation sites: cg06249499, cg06930016, cg11811655, cg13845406 |
| rs5953026 X:47317317 (G/A) | 0.602 (Combined, White) | FVC (increased) | HC = 2923 0.04 (0.01) $P = 1.9 \times 10^{-8}$ | 3′ of *ZNF157* | *Colocalization* | Colocalized gene expression trait: *RGN* |
| rs184101688 7:7140556 (C/A) | 0.001 (Combined, All) | FEV$_1$/FVC (decreased) | HC = 41 −9.42 (1.66) $P = 1.3 \times 10^{-8}$ | 5′ of *C1GALT1* | Overlap with known GWAS region | Previous GWAS variant* rsid, Chr:Pos (effect/other allele): rs4318980, 7:7216859 (A/G) EAF (discovery population): 0.415 (UK Biobank European ancestry) Trait (direction): FEV1/FVC (decreased) |
| rs74469188 16:81611365 (C/T) | 0.150 (COPD-enriched, African American) | FVC (increased) | HC = 714 0.14 (0.03) $P = 2.3 \times 10^{-8}$ | *CMIP* intronic | Association in UK Biobank with inconsistent direction of effect | UK Biobank European ancestry** EffHC = 64854 Beta (SE) = −0.017 (0.004) $P$-value = $1.4 \times 10^{-5}$ |
| rs371740347 1:196989333 (C/T) | 0.006 (COPD-enriched, All) | FVC (increased) | HC = 104 0.38 (0.07) $P = 1.1 \times 10^{-8}$ | *CFHR5* intronic | Phenome-wide association evidence with inconsistent direction of effect | Phenome-wide association*** trait (direction): respiratory failure, insufficiency, arrest (increased) $P$-value = $6.9 \times 10^{-5}$ |

Variants are reported based on genome-wide significance threshold of $P = 5 \times 10^{-8}$. All variant positions are presented based on Human Genome Build 38; EAF = effect allele frequency; HC = heterozygosity count; EffHC = effective heterozygosity count Genetic variant effects (betas) in TOPMed are reported for phenotypes under the heterogeneous variance model[56] such that the effect estimates reflect the scale of variance for FEV$_1$ (in L), FVC (in L) and FEV$_1$/FVC ratio (in %). $P$-values for genetic association in TOPMed as reported based on the SAIGE score test[55].
*Information on prior GWAS association reported based on result from Shrine et al.[17].
**Genetic variant effects for association in UK Biobank are reported for inverse normal transformed residualized lung function traits, following the model used in Shrine et al.[17]. $P$-values are reported based on the BOLT-LMM genetic association test[66].
***The phenome-wide association $P$-value is reported based on the SAIGE score test[55].

region, and the conditional association signal for severe COPD at rs9920270 near the previously reported *CHRNA3/5* signal was also colocalized with a methylation site in MESA (Supplementary Data 21).

We did not identify colocalization of any of our novel WGS signals with eQTLs from PBMCs in MESA.

**Overlap with pathways previously identified by GWAS.** Novel lung function-related genes identified by our study through colocalization with eQTL and overlap of novel variants with introns were represented in several pathways implicated by previous GWAS of lung function[17] (Supplementary Data 22). For example, the colocalized genes *MPC1*, *RGN*, and *NDUFB11* were represented in the phosphorus metabolic process. The genes *KANK1* and *CDK5RAP2* containing novel intronic variants were represented in the cytoskeleton organization and organelle organization pathways.

## Discussion

In this first pooled, multi-ethnic WGS analysis of pulmonary function and COPD from the NHLBI TOPMed Program, we identified at genome-wide significance 10 known GWAS loci and 22 distinct novel loci. We found evidence of replication with consistent direction of effect for four common variants, supportive evidence through colocalization for two additional signals, and a rare variant of large effect on reduced lung function (FEV$_1$/FVC ratio) overlapping a previously reported GWAS signal in the region of *C1GALT1*[17]. In gene-based analysis of putative LOF variants, we found an association with increased FEV$_1$/FVC in individuals with rare LOF variants for *ARHGEF17*, which has been reported previously as a candidate gene underlying pulmonary GWAS signals based on eQTL evidence and linkage disequilibrium with a nonsynonymous variant[17]. These results represent the existence of rare disease-associated variants residing within the region of previously reported common variant GWAS signals, and provide evidence of the possible causal gene and direction of effect.

Among the novel single variant associations identified, we found four distinct variants demonstrating evidence of replication with consistent directions of effect in analysis of pulmonary function for 321,047 European ancestry participants from the UK Biobank. These were rs17308514 near *PIAS1*, rs12556310 and rs35917906 within the region of *RGN*, and rs7188378 within the second intron of *FTO*. All of these replicated signals reflect common variants with minor allele frequencies greater than 0.3 in the discovery cohorts. We note that another *FTO* region common variant rs35420030 was reported in the recent GWAS of pulmonary function from the UK Biobank[17], however, the *FTO* region variant rs7188378 is considered novel in the current study, because the two variants have very different allele frequencies and are not in LD. Loss of *Fto* in mice has been shown to lead to reduced adipose tissue and lean body mass, as a result of increased energy expenditure[29], and *FTO* was also the first recognized RNA demethylase[30,31]. While neither our study nor the prior UK Biobank GWAS of lung function could assign a candidate gene to the *FTO* region variants based on colocalization with eQTL or other approaches, studies have linked obesity-related *FTO* variants to expression of the distal genes *IRX3* and *IRX5*[32]. In follow-up analysis of smoking behavior in the UK BIobank for the *PIAS1*-region variant rs17308514, we found that allele G, associated with decreased FVC, was also associated with decreased smoking initiation. Thus, while the same variant was associated with both FVC and smoking initiation, it does not appear that the association with FVC was mediated by smoking behavior.

The observed effect estimates of these replicated variants were modest, on the order of ~0.03 to ~0.06 standard deviations per copy of the observed effect allele in each case. In contrast, the rare variants rs184101688 (5′ of *C1GALT1*) and rs371740347 (intronic to *CFHR5*) demonstrated strikingly large effects, on the order of ~0.4 to ~0.9 standard deviations for each copy of the effect allele, a substantial effect in the context of common variant effects typically observed for single variants in genome-wide association studies[17]. In terms of the directions of effect, the rare non-coding *C1GALT1* variant rs184101688 is associated with reduced lung

function (FEV$_1$/FVC ratio), while the rare *CFHR5* variant is associated with increased FVC. In another example, rare pLOF variants in *ARHGEF17* were associated with increased FEV$_1$/FVC ratio as well as nominally associated with FEV$_1$, suggesting loss of function in this gene leads to improved lung function (increased FEV$_1$) rather than pulmonary restriction (reduced FVC). Additional studies will be needed to better understand the large effects of these rare variants.

For rs74469188 (intronic to *CMIP*), we observed a statistically significant association with a discordant direction of effect between TOPMed and UK Biobank, consistent with a false positive initial association. However, we cannot rule out the possibility that the discordance reflects a difference in race/ethnic- and/or disease status-specific effects[33] since the association with rs7449188 was identified in TOPMed WGS analysis of FVC in COPD-enriched African Americans while the replication was in a population-based sample of European ancestry participants from UK Biobank. Notably, the direction of effect for rs7449188 observed in African Americans from ARIC and MESA was consistent with that seen in the COPD-enriched African Americans, while the opposite direction of effect was observed with all of the White subgroups from TOPMed. It is likely that heterogeneity in genetic effects with respect to race/ethnicity and population-based versus COPD-enriched samples further hindered our replication efforts for other variants identified primarily in African Americans and/or COPD-enriched samples. This limitation highlights the need to prioritize recruitment of additional African Americans and COPD-enriched samples for future studies.

Integration of our WGS association results for pulmonary function and COPD with eQTL results from GTEx revealed some clues into the genes underlying some of the identified associations. In particular, for a few variants located at Xp11.3, we found the WGS association with FVC colocalization with eQTLs for the nearby genes *RGN*, *RNU6-1189P*, *USP11* and *NDUFB11*. Among these, only *RGN* was colocalized using eQTL from GTEx lung. Regucalcin, encoded by *RGN*, also known as senescence marker protein 30 (SMP30), is a highly conserved protein involved in calcium homeostasis, apoptosis, and oxidative stress. Human studies suggest that it plays a role in carcinogenesis[34], including lung cancer[35]. Animal studies have linked regucalcin to aging, due to its age associated down-regulation[36]. Smp30 knockout (Smp30$^{Y/-}$) mice developed increased lung cell apoptosis and emphysema in response to cigarette smoke, and this effect was attenuated by vitamin C[37,38]. *RGN* was also included in multiple pathways previously implicated by GWAS studies, including protein kinase activity, phosphorus metabolic process, and phosphotransferase activity. Our findings lend human evidence to the importance of *RGN* in COPD, underscoring the role of aging in pathogenesis of the disease[39].

The signal at rs17308514 near the gene *PIAS1* was colocalized with one methylation site within the region, providing additional evidence supporting this signal for which we also observed replication in UK Biobank. *PIAS1* has been shown to be phosphorylated in response to pro-inflammatory stimuli[40], its expression is reported to be associated with risk and survival for several chronic diseases including cancer[41,42] and multiple sclerosis[43].

For the variant rs9295345, we found it colocalized with expression of *MPC1*, *PRR18*, as well as four correlated methylation sites within the region. For one of these methylation traits, cg06249499, we also found that increased levels of measured methylation were associated with increased FEV$_1$ among MESA participants. *MPC1* (mitochondrial pyruvate carrier 1) plays a role in transport of pyruvate into mitochondria and down regulation of *MPC1* has been shown to accelerate progression of lung

adenocarcinoma[44]. Like *RGN*, MPC1 is also a member of the phosphorus metabolic process gene ontology term, suggesting further study of this pathway is warranted in relation to COPD.

For seven of the newly associated regions, the most strongly associated variants were common autosomal variants with MAF greater than 10%. Two of these novel common variant associations were identified only in stratified analysis of African Americans. Among these, the *GALNT18* region variant rs4076943 identified in analysis of COPD-enriched African Americans was previously reported for a suggestive association ($P = 2.4 \times 10^{-7}$) with post-bronchodilator FEV$_1$ in COPDGene African Americans[45]. The remaining five common autosomal associations (in or near *RPS6KA2*, *KANK1*, *PIAS1*, *FTO*, and *LRP1B*) were identified in analyses that included COPD-enriched samples, for whom there have not been many prior multi-study GWAS efforts examining quantitative PFT traits. Further, we observed that many of the WGS associations we identified had stronger effects in the COPD-enriched cohorts compared to population-based cohorts, including the *FTO* region variant rs7188378. While this observation may indicate differences in the underlying genetic association effects with respect to smoking exposures and disease status, it is also possible that these results reflect differences in ascertainment and smoking exposures in the COPD-enriched cohorts. Perhaps due to these particular features of our study sample, many of our other novel WGS associated variants did not replicate in analysis of the population-based European ancestry samples from the UK Biobank. Additionally, we attempted replication in African-ancestry and Hispanic-American samples, but did not confirm our findings in those race/ethnic groups, likely due in part to the smaller sample sizes available for replication in diverse populations.

The most strongly associated variants for nine of the newly reported associated regions were rare/infrequent variants with MAF less than 1%. In examination of imputed genotypes from MESA, a representative cohort including all of the major race/ethnic groups that were part of the current investigation, several of these variants did not show up in the imputation results based on 1000 Genomes[20,46] and the Haplotype Reference Consortium[47]. Among those variants that did have imputed genotypes available, the R-squared values between genotypes from imputation versus TOPMed sequencing were relatively poor for these rare/infrequent variants, suggesting that the set of variants with newly reported associations for PFT and COPD in this study were not well-covered by prior GWAS efforts. We note that most of these novel rare variants were imputed successfully using the TOPMed reference panel, suggesting future GWAS efforts may benefit from high quality imputation resulting from this largest multi-ethnic reference panel to date.

Our study used pre-bronchodilator pulmonary function from a broad range of cohorts. We leveraged extensive quality control and harmonization efforts for pulmonary function using standardized criteria[48], and previously demonstrated that the effect of using pre- vs post-bronchodilator measures has minimal impact on genetic association[49]. Our analysis of cross-sectional data, adjusting for age and smoking exposure, does not specifically address determinants of change in lung function over time or other phenotypes that may be important in COPD. While we tested whether our associations were explained by cigarette smoking, whether the effects of these variants may be mediated by, or affected by, other environmental pathways is not known. Prior studies of gene by environment interaction with smoking suggest that such studies will require accurate phenotypic measurements in very large sample sizes[50,51]. We did not examine the relationship of our identified variants with environmental factors including occupational exposures and air pollution, in part due to the lack of clean harmonized data on these exposures across the

full set of cohorts in our study. As with prior genetic association studies, the specific variants identified account for a small fraction of the heritability of COPD[18]. While some of the rare variants identified in our study demonstrate increased effect sizes, their impact at a population level is offset by their low allele frequency.

In conclusion, we report on the largest whole genome sequencing effort to date to identify genetic loci linked to lung function traits and COPD. Our study demonstrates the value of WGS approaches, particularly for identification of associations with variants that may be difficult to impute, including rare and infrequent variants, common variants in African Americans, and X chromosome variants that have been excluded from many of the prior published GWAS studies. In addition, our study provides evidence that some previously reported GWAS loci may overlap with gene-based associations with putative LOF variants. Our study's inclusion of COPD cases may have increased our ability to identify variants that have stronger effects in disease-ascertained cohorts, with the caveat that COPD-enriched cohorts may also be subject to bias due to ascertainment. Limitations of our study include small sample sizes for replication in non-European ancestry populations or in cohorts representing COPD-enriched cases. Future efforts will include expanded WGS analyses to leverage forthcoming genome-wide imputation using TOPMed as a reference panel, which is expected to provide substantially improved imputation quality compared to existing reference panels for both European and non-European ancestry samples. In addition, as additional 'omics data are generated through TOPMed and other sources, we intend to leverage those data to further inform the molecular mechanisms underlying these genetic associations.

## Methods

**Study samples**. Participants were included from six population- and family-based cohorts (the Atherosclerosis Risk in Communities [ARIC] Study, the Cleveland Family Study [CFS], the Cardiovascular Health Study [CHS], the Framingham Heart Study [FHS], the Jackson Heart Study [JHS], and the Multi-Ethnic Study of Atherosclerosis [MESA]) and two COPD-enriched studies (the Genetic Epidemiology of COPD [COPDGene] Study, which enrolled cases, controls, and additional smokers with varied lung function; and the family-based Boston Early Onset COPD [EOCOPD] Study, from which we sequenced unrelated probands). All participants gave informed consent and the institutional review boards at the University of Virginia, Brigham and Women's Hospital and all participating centers approved the study, Detailed cohort descriptions are provided in the Supplementary Methods.

**Phenotype definition**. Phenotype harmonization of Pulmonary Function Test (PFT) measures, including pre-bronchodilator $FEV_1$, FVC, and $FEV_1/FVC$ ratio, was conducted following the protocol of the NHLBI Pooled Cohorts Study[48]. For studies with multiple time points, we worked with investigators from each study to determine the most practical way to construct a cross-sectional subset of data (Supplementary Methods). All spirometry data utilized for this effort were obtained as pre-bronchodilator measures. Based on the quantitative measures of PFT and self-reported categories of race/ethnicity, we calculated race/ethnic-specific predicted values of $FEV_1$ for White, African American, and Hispanic participants using the equations of Hankinson[52] that were determined for White, African American, and Mexican American reference populations, respectively. For Asian participants, we used the Hankinson equations determined for White, and then multiplied by a reduction factor of 0.88[53]. COPD cases and controls were then defined as follows:

- Moderate-to-Severe COPD: pre-bronchodilator $FEV_1 < 80\%$ predicted and $FEV_1/FVC < 0.7$,
- Severe COPD: pre-bronchodilator $FEV_1 < 50\%$ predicted and $FEV_1/FVC < 0.7$, and
- Controls: pre-bronchodilator $FEV_1 \geq 80\%$ predicted and $FEV1/FVC \geq 0.7$.

**Whole genome sequence data**. Whole Genome Sequencing (WGS) in TOPMed had, on average, deep (~30X) coverage with joint-sample variant calling and variant level quality control in >50,000 TOPMed samples (freeze 5b)[54]. Additional details regarding quality control of genotype data for the present analyses are included in the Supplementary Methods.

**Single variant analyses**. Analyses were conducted using SAIGE-LMM v0.29.4.4[55] and stratified by study design (population- and family-based studies vs. COPD-enriched studies), as well as combined. Within strata, separate analyses in Whites vs. African Americans, as well as pooled across race/ethnic groups were undertaken.

*Quantitative trait analysis of $FEV_1$, FVC and $FEV_1/FVC$*: We incorporated covariate adjustment for age, age[2], sex, height, height[2], weight (FVC only), study, current smoking, former smoking, pack-years of smoking, first 10 principal components (PCs) of ancestry, and sequencing center. With these covariates, we implemented a heterogeneous variance model[56] to account for different phenotype distributions across studies. To do so, cohort-specific residuals were obtained after adjustment for the stated covariates using a linear mixed model, implemented in R v3.5.2/GENESIS v2.12.2[57]. Phenotypes for pooled analyses were then constructed by applying inverse normal transform to the cohort-specific residuals, and then scaling these residuals by their cohort-specific variance. Accordingly, the scale of the beta estimates obtained by quantitative trait analysis corresponds approximately to the original scale on which the traits were measured.

*Dichotomous trait analysis of COPD*: Case-control analyses incorporated covariate adjustment for age, sex, study, pack-years, and ever vs. never smoking, first 10 PCs of ancestry and sequencing center.

*Variant-level filter*: In addition to standard quality control filters applied to the TOPMed Freeze 5b data set[54], whole genome sequence analysis results were filtered on heterozygosity count (HC) > 30 for quantitative trait analyses and expected HC > 30 among cases for case-control analyses. We applied a genome-wide significance threshold of $P < 5 \times 10^{-8}$ for reporting novel and known variants in this manuscript.

*Identification of novel versus previously reported variants*: Variants previously documented in the CHARGE/SpiroMeta[13,14,16], UK BiLEVE[15], or UK Biobank[17] GWAS of pulmonary function, as well as the ICGC[49] and ICGC/UK Biobank combined[18] GWAS of COPD were considered known prior to the current WGS analysis. Additionally, quantifying linkage disequilibrium (LD) based on our TOPMed European ancestry samples, those variants demonstrating LD R-squared > 0.2 with one or more previously reported GWAS variants within a +/− 5 Mb window were considered known. The remaining variants that were not in LD with known GWAS variants or were located beyond 5 Mb from the lead variants for known loci were considered novel in the current study.

*Annotation of novel variants*: Novel variants identified by WGS analyses were annotated using the WGS Annotator (WGSA) v0.7[58].

**Conditional analysis based on WGS analysis**. We assessed dependence of signals using GCTA-COJO v1.93.2[59], a summary statistics-based approach for conditional analysis. The analysis was focused on variants within a +/− 1-Mb region of the most strongly associated known and novel signals. Since the MHC region has more complex structure, we conducted the analysis on a rough 9-Mb region for the variants on chromosome 6 (region chr6:27,000,000 − 36,000,000 for chr6:32,167,360 and chr6:32,182,024). Conditional analysis was conducted around 34 novel and known top associated variants. In total, the region examined had length $33*2 + 9 = 75$ Mb, which consisted of roughly 2.5% of the human genome $(75/3000 = 2.5\%)$. Based on the Bonferroni correction, any variant that had conditional $P < 2 \times 10^{-6}$ was designated a potentially distinct signal $(P < 5 \times 10^{-8}/0.025)$.

**Comparison of TOPMed WGS calls with GWAS imputed genotypes**. To examine how well the novel associated variants were represented by prior GWAS efforts, we computed the R-squared of genotypes using variant calls from TOPMed Freeze 5b compared to genotypes obtained using imputation of genome-wide SNP genotyping arrays in MESA to various reference panels including the 1000 Genomes Phase 1[20], 1000 Genomes Phase 3[46] and the Haplotype Reference Consortium (HRC)[47]. Details are provided in the Supplementary Methods.

**Rare putative loss of function (pLOF) variant burden test**. A gene-based burden test was conducted on 228,966 pLoF variants. These variants were previously identified using Loss-Of-Function Transcript Effect Estimator (LOFTEE) v0.3-beta[60,61] and Variant Effect Predictor (VEP) v94[62]. Variants used in our analysis included stop-gained, frameshift, and splice site disturbing variants. Only SNPs with minor allele frequency (MAF) of <0.5% were included. In total, we observed pLOF variants in 17,142 genes that were defined based on GENCODE v29. We conducted burden tests to examine the association between quantitative lung function traits and gene burden using SAIGE-GENE v0.36.3.3[63]. Gene-level burden was generated by aggregating low frequency pLoF variants weighted by their allele frequencies. For reporting significant signals, the results were filtered on cumulative minor allele count > 5. We applied different significance thresholds for three classes of genes under consideration:

(1) *Genome-wide screen*: We used a Bonferroni-corrected significance threshold of $P < 0.05/17,142$ genes $= 2.9 \times 10^{-6}$ in examining all genome-wide genes by pLOF analysis.

(2) *Mendelian candidate genes*: The Bonferroni-corrected significance threshold was derived as $P < 0.05/25$ genes $= 2.0 \times 10^{-3}$ to account for examination of 25 candidate Mendelian genes, selected for their relevance to COPD and emphysema, cutis laxa and the telomerase pathway (listed in Supplementary Table 3)[64,65].

(3) *Genes overlapping GWAS regions:* We examined candidate genes that overlapped with the 100 kb flank regions of the GWAS top associated variants. In total, we checked 68 genes for $FEV_1$, 78 genes for FVC, 95 genes for $FEV_1/FVC$, and 92 genes for COPD (see Supplementary Data 23-26). The Bonferroni-corrected significance thresholds were derived as 0.05 divided by the number of genes being tested: $7.4 \times 10^{-4}$ for $FEV_1$, $6.4 \times 10^{-4}$ for FVC, $5.3 \times 10^{-4}$ for $FEV_1/FVC$ and $5.4 \times 10^{-4}$ for COPD.

**Replication cohorts and analysis.** For those variants demonstrating novel associations with one or more measures of pulmonary function or COPD, we examined evidence of replication in the UK Biobank[17] and the Hispanic Community Health Study/Study of Latinos (HCHS/SOL)[27]. Only variants passing quality control and other filters for analyses in the respective replication cohorts were considered when we performed multiple comparisons corrections to determine which variants demonstrated evidence of replication. Details provided in the Supplementary Methods.

**Phenome-wide association analysis.** Among the novel WGS variants identified in our study, we carried out phenome-wide association study (PheWAS) analysis for autosomal variants passing filter on effective HC > 30 and imputation info score > 0.3 using the HRC for imputation in 408,961 White British participants from the UK Biobank. For each of these variants, we carried out PheWAS across 1403 binary phenotypes reported in the UK Biobank, which were constructed from composites of ICD-9 and ICD-10 codes, with results available publicly through the UK Biobank ICD PheWeb (http://pheweb.sph.umich.edu/SAIGE-UKB/)[55]. Results for specific traits were further filtered based on expected HC > 30 in cases. Among the 1403 traits examined, 71 were respiratory disease traits. We used different suggestive thresholds for significant p-values based on Bonferroni correction for respiratory diseases and other diseases: 0.05 divided by 71 ($P < 7.0 \times 10^{-4}$) and 1,332 ($P < 3.8 \times 10^{-5}$).

**Molecular QTL colocalization analysis and follow-up.** We examined colocalization with molecular QTLs of gene expression (eQTL) and methylation (mQTL): eQTLs in 48 tissues from GTEx v7, eQTLs in PBMCs in MESA, and mQTLs in whole blood in MESA -within 500 kb of the lead WGS variants (novel variants in Supplementary Data 2, as well as those identified from conditional analysis in Supplementary Data 11) using Bayesian colocalization as implemented in R/coloc. v3.1[28]. We report the results where the model of a single shared causal variant driving both associations signals (PP4) is strongly preferred over a model of two distinct causal variants (PP3)—PP4/(PP3 + PP4) ≥ 0.9. We require adequate power for these results to detect colocalization—PP3 + PP4 ≥ 0.8. Those methylation sites demonstrating colocalization with WGS signals were followed up to examine association of measured methylation with corresponding lung function traits in MESA, and results are presented after Bonferroni correction for the number of colocalized methylation sites. Details are provided in the Supplementary Methods.

**Overlap with pathways previously implicated by GWAS.** For genes implicated by eQTL colocalization, as well as other selected genes, we examined their overlap with pathways previously implicated by GWAS studies. Details are provided in the Supplementary Methods.

**Reporting summary.** Further information on research design is available in the Nature Research Reporting Summary linked to this article.

## Data availability

Individual whole-genome sequence data for TOPMed whole genomes are available through dbGaP. The dbGaP accession numbers are: Atherosclerosis Risk in Communities (ARIC) phs001211, Cardiovascular Health Study (CHS) phs001368, Cleveland Family Study (CFS) phs000954, Framingham Heart Study (FHS) phs000974, Jackson Heart Study (JHS) phs000964, Multi-Ethnic Study of Atherosclerosis (MESA) phs001416, Boston Early-Onset COPD (EOCOPD) phs000946, and Genetic Epidemiology of COPD (COPDGene) phs000951. Data in dbGaP can be downloaded by controlled access with an approved application submitted through their website: https://www.ncbi.nlm.nih.gov/gap.

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

## Acknowledgements

Whole genome sequencing (WGS) for the Trans-Omics in Precision Medicine (TOPMed) program was supported by the National Heart, Lung and Blood Institute (NHLBI). WGS for "NHLBI TOPMed: Atherosclerosis Risk in Communities (ARIC)" (phs001211) was performed at the Baylor College of Medicine Human Genome Sequencing Center (HHSN268201500015C and 3U54HG003273-12S2) and the Broad Institute for MIT and Harvard (3R01HL092577-06S1). WGS for "NHLBI TOPMed: Cardiovascular Health Study (CHS)" (phs001368) was performed at the Baylor College of Medicine Human Genome Sequencing Center (HHSN268201500015C). WGS for "NHLBI TOPMed: The Cleveland Family Study" (phs000954) was performed at the University of Washington Northwest Genomics Center (3R01HL098433-05S1). WGS for "NHLBI TOPMed: Whole Genome Sequencing and Related Phenotypes in the Framingham Heart Study" (phs000974) was performed at the Broad Institute of MIT and Harvard (HHSN268201500014C and 3R01HL092577-06S1). WGS for "NHLBI TOPMed: The Jackson Heart Study" (phs000964) was performed at the University of Washington Northwest Genomics Center (HHSN268201100037C). WGS for "NHLBI TOPMed: Multi-Ethnic Study of Atherosclerosis (MESA)" (phs001416) was performed at the Broad Institute of MIT and Harvard (3U54HG003067-13S1 and HHSN268201500014C). WGS for "NHLBI TOPMed: Boston Early-Onset COPD Study in the TOPMed Program" (phs000946) was performed at the University of Washington Northwest Genomics Center (3R01HL089856-08S1). WGS for "NHLBI TOPMed: Genetic Epidemiology of COPD (COPDGene) in the TOPMed Program" (phs000951) was performed at the University of Washington Northwest Genomics Center (3R01 HL089856-08S1) and the Broad Institute of MIT and Harvard (HHSN268201500014C). Centralized read mapping and genotype calling, along with variant quality metrics and filtering were provided by the TOPMed Informatics Research Center (3R01HL-117626-02S1; contract HHSN268201800002I). Phenotype harmonization, data management, sample-identity QC, and general study coordination, were provided by the TOPMed Data Coordinating Center (3R01HL-120393-02S1; contract HHSN268201800001I). The TOPMed MESA Multi-Omics project was conducted by the University of Washington and LABioMed (HHSN268201500003I/HHSN26800004). Phenotype harmonization for pulmonary traits was contributed by the NHLBI Pooled Cohorts Study with funding from NIH/NHLBI R21 HL121457, R21 HL129924, K23 HL130627, R01 HL077612. This research was supported by a TOPMed Analysis Support Award (through U01 HL117626), NIH/NHLBI R01 HL131565 (A.M.), R01 HL142028 (T.L.), R01 HL135142; R01 HL137927; R01 HL089856; and R01 HL147148 (M.H.C.), and K01-HL129039 (D.Q.). Study-specific acknowledgments are given in Supplementary Notes 1–3. We gratefully acknowledge the studies and participants who provided biological samples and data for TOPMed. The views expressed in this manuscript are those of the authors and do not necessarily represent the views of the National Heart, Lung, and Blood Institute; the National Institutes of Health; or the U.S. Department of Health and Human Services. A full list of authors for the NHLBI Trans-Omics for Precision Medicine (TOPMed) Consortium is provided at https://www.nhlbiwgs.org/topmed-banner-authorship and in Supplementary Note 4.

## Author contributions

R.G.B., E.K.S., A.M., M.H.C., G.T.O., S.S.R., S.J.L. and G.R.A., designed the study. X.Z., D.Q., C.Y., S.K., W.K., Y.M., N.S., C.B., S.A.G.T., P.S., P.P.B., D.P., B.Y., L.A.L., J.D., B.E.C., J.L., S.A.G., M.D., I.R., L.A.C., L.R.L., T.M.B., A.C.M., B.M.C., R.S.V., J.G.W., K.D.T., P.D., W.C.J., E.C., X.G., Y.L., R.P.T., K.G.A., F.A., D.J.V., G.J.P., J.I.R., K.C.B., D.J., D.A.N., D.M.M., G.A.M., H.D., S.D.-P., N.D., S.G., S.S.R., G.T.O., S.R., R.M.R., C.C.L., M.L.D., L.K.P., K.M.B., R.M.K., L.V.W., M.D.T., S.J.L., T.L., E.C.O., G.R.A., E.K.S., R.G.B., M.H.C. and A.M. acquired, analyzed or interpreted data. X.Z., D.Q., M.H.C. and A.M. wrote the manuscript. X.Z., P.P.B., C.A.L., C.C.L., E.C.O., G.R.A., M.H.C., A.M., the NHLBI Trans-Omics for Precision Medicine (TOPMed) Consortium, and the TOPMed Lung Working Group provided administrative, technical or material support. All authors contributed to critical revision of the manuscript.

## Competing interests

In the past three years, Edwin K. Silverman and Michael H. Cho have received grant and travel support from GlaxoSmithKline and Bayer. Michael H. Cho has received consulting and speaking fees from Illumina and AstraZeneca. Tuuli Lappalainen is a scientific advisory board member of Variant Bio with equity and Goldfinch Bio. Bruce M. Psaty serves on the Steering Committee of the Yale Open Data Access Project funded by Johnson & Johnson. Goncalo R. Abecasis is an employee of Regeneron Pharmaceuticals and owns stock and stock options for Regeneron Pharmaceuticals. All other authors have no competing interests to declare.

## Additional information

Xutong Zhao[1], Dandi Qiao[2], Chaojie Yang[3], Silva Kasela[4,5], Wonji Kim[2], Yanlin Ma[3], Nick Shrine[6], Chiara Batini[6], Tamar Sofer[7,8], Sarah A. Gagliano Taliun[1], Phuwanat Sakornsakolpat[2], Pallavi P. Balte[9], Dmitry Prokopenko[2], Bing Yu[10], Leslie A. Lange[11], Josée Dupuis[12], Brian E. Cade[7,8], Jiwon Lee[8], Sina A. Gharib[13], Michelle Daya[11], Cecelia A. Laurie[14], Ingo Ruczinski[15], L. Adrienne Cupples[12,16], Laura R. Loehr[17], Traci M. Bartz[14], Alanna C. Morrison[18], Bruce M. Psaty[19,20], Ramachandran S. Vasan[16,21], James G. Wilson[22], Kent D. Taylor[23], Peter Durda[24], W. Craig Johnson[14], Elaine Cornell[24], Xiuqing Guo[23], Yongmei Liu[25], Russell P. Tracy[24], Kristin G. Ardlie[26], François Aguet[26], David J. VanDenBerg[27], George J. Papanicolaou[28], Jerome I. Rotter[23], Kathleen C. Barnes[11], Deepti Jain[14], Deborah A. Nickerson[29], Donna M. Muzny[30], Ginger A. Metcalf[30], Harshavardhan Doddapaneni[30], Shannon Dugan-Perez[30], Namrata Gupta[26], Stacey Gabriel[26], Stephen S. Rich[3], George T. O'Connor[31], Susan Redline[7,8,32], Robert M. Reed[33], Cathy C. Laurie[14], Martha L. Daviglus[34], Liana K. Preudhomme[35], Kristin M. Burkart[9], Robert C. Kaplan[36,37], Louise V. Wain[6,38], Martin D. Tobin[6,38], Stephanie J. London[39], Tuuli Lappalainen[4,5], Elizabeth C. Oelsner[9], Goncalo R. Abecasis[1], Edwin K. Silverman[2], R. Graham Barr[9], NHLBI Trans-Omics for Precision Medicine (TOPMed) Consortium, TOPMed Lung Working Group, Michael H. Cho[2,40]✉ & Ani Manichaikul[3,40]✉

[1]Center for Statistical Genetics, and Department of Biostatistics, University of Michigan, Ann Arbor, MI 48109, USA. [2]Channing Division of Network Medicine, Department of Medicine, Brigham and Women's Hospital and Harvard Medical School, Boston, MA 02115, USA. [3]Center for Public Health Genomics, University of Virginia, Charlottesville, VA 22908, USA. [4]New York Genome Center, New York, NY 10013, USA. [5]Department of Systems Biology, Columbia University, New York, NY 10032, USA. [6]Genetic Epidemiology Group, Department of Health Sciences, University of Leicester, Leicester LE1 7RH, United Kingdom. [7]Department of Medicine, Harvard Medical School, Boston, MA 02115, USA. [8]Division of Sleep and Circadian Disorders, Brigham and Women's Hospital, Boston, MA 02115, USA. [9]Department of Medicine, Columbia University Medical Center, New York, NY 10032, USA. [10]Department of Epidemiology, Human Genetics & Environmental Sciences, UTHealth School of Public Health, Houston, TX 77030, USA. [11]Division of Biomedical Informatics and Personalized Medicine, Department of Medicine, University of Colorado School of Medicine Anschutz Medical Campus, Aurora, CO 80045, USA. [12]Department of Biostatistics, Boston University School of Public Health, Boston, MA 02118, USA. [13]Division of Pulmonary, Critical Care and Sleep Medicine, University of Washington, Seattle, WA 98109, USA. [14]Department of Biostatistics, University of Washington, Seattle, WA 98195, USA. [15]Department of Biostatistics, Johns Hopkins Bloomberg School of Public Health, Baltimore, MD 21205, USA. [16]Boston University and the National Heart Lung and Blood Institute's Framingham Heart Study, Framingham, MA 01702, USA. [17]Department of Medicine, UNC School of Medicine, Chapel Hill, NC 27599, USA. [18]Human Genetics Center, Department of Epidemiology, Human Genetics, and Environmental Sciences, School of Public Health, The University of Texas Health Science Center at Houston, Houston, TX 77030, USA. [19]Cardiovascular Health Research Unit, Departments of Medicine, Epidemiology, and Health Services, University of Washington, Seattle, WA 98101, USA. [20]Kaiser Permanente Washington Health Research Institute, Seattle, WA 98101, USA. [21]Department of Preventive Medicine and Epidemiology, Boston University School of Medicine and Public Health, Boston, MA 02118, USA. [22]Department of Physiology and Biophysics, University of Mississippi Medical Center, Jackson, MS 39216, USA. [23]The Institute for Translational Genomics and Population Sciences, The Department of Pediatrics, The Lundquist Institute for Biomedical Innovation at Harbor-UCLA Medical Center, Torrance, CA 90502, USA. [24]Department of Pathology and Laboratory Medicine, Robert Larner, M.D. College of Medicine, University of Vermont, Burlington, VT 05405, USA. [25]Department of Medicine, Division of Cardiology, Duke Molecular Physiology Institute, Duke University Medical Center, Durham, NC 27701, USA. [26]Broad Institute of MIT and Harvard, Cambridge, MA 02142, USA. [27]Department of Preventive Medicine, University of Southern California, Los Angeles, CA 90033, USA. [28]Division of Cardiovascular Sciences, National Heart, Lung, and Blood Institute, National Institutes of Health, Bethesda, MD 20892, USA. [29]Department of Genome Sciences, University of Washington, Seattle, WA

98195, USA. [30]The Human Genome Sequencing Center, Baylor College of Medicine, Houston, TX 77030, USA. [31]Boston University School Of Medicine, Pulmonary Center, Boston, MA 02118, USA. [32]Division of Pulmonary, Critical Care, and Sleep Medicine, Beth Israel Deaconess Medical Center, Boston, MA 02215, USA. [33]Division of Pulmonary and Critical Care Medicine, University of Maryland School of Medicine, Baltimore, MD 21201, USA. [34]Institute for Minority Health Research, University of Illinois at Chicago, Chicago, IL 60612, USA. [35]Department of Psychology, University of Miami, Miami, FL 33124, USA. [36]Department of Epidemiology and Population Health, Albert Einstein College of Medicine, Bronx, New York, NY 10461, USA. [37]Public Health Sciences Division, Fred Hutchinson Cancer Research Center, Seattle, WA 98109, USA. [38]National Institute for Health Research, Leicester Respiratory Biomedical Research Centre, Glenfield Hospital, Leicester LE3 9QP, UK. [39]Epidemiology Branch, National Institute of Environmental Health Sciences, National Institutes of Health, Department of Health and Human Services, Durham, NC 27709, USA. [40]These authors jointly supervised this work: Michael H. Cho and Ani Manichaikul. ✉email: remhc@channing.harvard.edu; amanicha@virginia.edu

## NHLBI Trans-Omics for Precision Medicine (TOPMed) Consortium

Goncalo R. Abecasis[1], François Aguet[26], Kristin G. Ardlie[26], Kathleen C. Barnes[11], R. Graham Barr[9], Brian E. Cade[7,8], Michael H. Cho[2,40✉], Elaine Cornell[24], L. Adrienne Cupples[12,16], Michelle Daya[11], Peter Durda[24], Stacey Gabriel[26], Xiuqing Guo[23], Namrata Gupta[26], Deepti Jain[14], W. Craig Johnson[14], Robert C. Kaplan[36,37], Wonji Kim[2], Leslie A. Lange[11], Cathy C. Laurie[14], Cecelia A. Laurie[14], Jiwon Lee[8], Yongmei Liu[25], Ani Manichaikul[3,40✉], Deborah A. Nickerson[29], George J. Papanicolaou[28], Bruce M. Psaty[19,20], Dandi Qiao[2], Ramachandran S. Vasan[16,21], Susan Redline[7,8,32], Robert M. Reed[33], Stephen S. Rich[3], Jerome I. Rotter[23], Ingo Ruczinski[15], Edwin K. Silverman[2], Tamar Sofer[7,8], Kent D. Taylor[23], Russell P. Tracy[24], David J. VanDenBerg[27] & James G. Wilson[22]

## TOPMed Lung Working Group

Pallavi P. Balte[9], Kathleen C. Barnes[11], R. Graham Barr[9], Brian E. Cade[7,8], Michael H. Cho[2,40✉], Josée Dupuis[12], Sina A. Gharib[13], Robert C. Kaplan[36,37], Silva Kasela[4,5], Wonji Kim[2], Leslie A. Lange[11], Tuuli Lappalainen[4,5], Jiwon Lee[8], Stephanie J. London[39], Ani Manichaikul[3,40✉], George T. O'Connor[31], Dmitry Prokopenko[2], Dandi Qiao[2], Susan Redline[7,8,32], Robert M. Reed[33], Stephen S. Rich[3], Ingo Ruczinski[15], Phuwanat Sakornsakolpat[2], Edwin K. Silverman[2], Kent D. Taylor[23], Chaojie Yang[3], Bing Yu[10] & Xutong Zhao[1]

Lists of authors and their affiliations appear in the Supplementary Information.

