## [Peer Review File · Nature Communications]

Reviewers' comments:

Reviewer #1 (Remarks to the Author):

The authors present a large-scale COPD and related phenotypes analysis that uses WGS data. Unfortunately, most of the significant associations that they present are just barely past the significance threshold and either do not replicate in another population or were not able to be attempted to replicate. It seems most likely to me that most of the novel hits presented here are not actual associations that will hold up.

Here are some things the authors could do to improve this problem with the manuscript:

1) Change the main table to include columns about replication so that the reader can clearly follow what happened.

2) Only call the ones that firmly replicate the novel hits. Don't prop up the ones that didn't replicate as possibilities unless you can show that the power was limited (although this seems unlikely since the replication sample size was so large). For anything that you can't get additional data for to replicate, list it as a possibility and not as a novel hit.

3) One variant is listed as "replicating" in the UKB with an opposite direction of effect. This can make sense as a hypothesis in a GWAS where all the variation was not directly genotyped, but here the original data are from WGS. If the original signal were driven by something else or combination of something else nearby, then you would be able to see it. If you are going to propose that the UKB data may reflect a replication of some other nearby signal that appears different due to LD and population structure differences, then you need to really investigate what is going on here and make a good case for it. To me, the most likely explanation is that this is a variant that is prone to false associations due to population structure.

4) For variants that weren't imputed in the UKB, try the exome data.

5) For a better study overall, use the UKB exome data in the discovery phase, especially for the rare variant collapsing test. While the current discovery datasets used WGS, the only parts missing from UKB exomes would be any non-coding variants too rare to be imputed, which are for the most part also too rare to have power for discovery in this study regardless.

6) The method of using allele counts in the cases to set the cutoff can lead to test statistic inflation. The recommended method to prevent false associations is to determine the allele frequency where one would expect at least x minor alleles in the smaller group (<http://www.nealelab.is/blog/2017/9/11/details-and-considerations-of-the-uk-biobank-gwas>). Also, one place in the paper indicates that your cutoff is 20, another place says 30.

Other major comments:

6) SAIGE-LMM wasn't originally made for gene-based collapsing analyses. The way its modeling worked could make collapsed results unreliable. A separate SAIGE-GENE software recently became available, but they explicitly stated in their paper that they didn't have it working for LMM yet. It looks like the most recent version may be able to do both collapsing and LMM though, so please confirm that you are using this latest version and that it is set up to do the test exactly as you have.

7) The way the results are presented is difficult to slog through. Please try to summarize more broadly when possible and put the fine details in tables or supplement. The results would also be easier to get through if they more clearly told a story.

Minor comments:

- 8) Provide information the power in this study.
- 9) Indicate whether there were any previously known GWAS hits that you missed.
- 10) Would like more details on how the ICD9 and 10 codes were combined.

Reviewer #2 (Remarks to the Author):

This is a well written and executed study of genetic variants in pulmonary function and COPD. The authors have assembled an extremely large number of multi-ethnic subjects across multiple studies in which whole genome sequencing has confirmed a number of loci previously identified in GWAS studies, as well as identifying novel loci that they attempt to replicate. Strengths of this effort include the number of African Americans studied, the whole genome approach that allows for more complete coverage of genetic variation, and the replication, co-localization, and eQTL efforts that provide support for their findings. Inclusion of COPD-enriched subjects provides additional power to evaluated genetic contributions to the extreme ends of the phenotype spectrum. There are some additional discussion points for consideration that are briefly mentioned in the discussion but deserve additional attention.

1. The replication datasets do not reflect the populations of the study populations. The numbers of African Americans in the replication dataset is relatively small with no emphasis on COPD-enriched participants as seen in the study population, and the number of Hispanics is large in comparison the study populations. The inability to replicate findings in some instances is likely related to these issues.
2. The phenotypes are not consistent across study populations, i.e., sometimes baseline PFTs are used, sometime most recent, some are pre- and post-bronchiodilator use. The potential for misclassification, or at least inconsistencies across study populations should be addressed.
3. It is unclear why variant associations with smoking initiation and cessation are tested in the replication sample. These are not associations that are reported for the study populations.
4. Some discussion of attributable risk associated with identified variants would provide perspective, as would a discussion of what clinical usefulness these variants might have.

Reviewer #3 (Remarks to the Author):

Zhao X et al. Whole genome sequence analysis of pulmonary function and COPD in 20,000 multi-ethnic participants from the NHLBI trans-omics for precision medicine (TOPMed) program. This is a well written manuscript where associations were observed between lung function and COPD with 32 loci, among those 22 were novel. Table 2 brings these observations together. However, how do we bring these observations under a wider biological or even clinical umbrella. ARHGEF17 variants leading to loss of function were associated with increased FEV1/FVC indicating either normal lung function or pulmonary restriction due to perhaps fibrosis. FTO has mentioned as a new factor without further explanations. This fat mass and obesity-associated protein also known as alpha-ketoglutarate-dependent dioxygenase FTO is an enzyme in humans encoded by the FTO gene located on chromosome 16. As one homolog in the AlkB family proteins, it is the first mRNA demethylase that has been identified. Regucalcin is another marker that stands out to be important for interstitial lung micro-environment. Taken all the associations together how much do they explain the occurrence of COPD. To what extent do they point to the most important pathways that may be deregulated by external factors, such as smoke, fine-particle dust. I would clarify and extend this issue in the discussion.

Response to Reviewers

We have revised the manuscript according to the suggestions of the three reviewers. We now highlight all changes in the text that reflect these critical points and suggestions in **blue**.

Reviewer #1 (Remarks to the Author):

The authors present a large-scale COPD and related phenotypes analysis that uses WGS data. Unfortunately, most of the significant associations that they present are just barely past the significance threshold and either do not replicate in another population or were not able to be attempted to replicate. It seems most likely to me that most of the novel hits presented here are not actual associations that will hold up.

We thank the Reviewer for the thoughtful comments. We agree with the Reviewer that it is better to focus the manuscript on those loci that can be supported in some form beyond our discovery effort. We provide our detailed responses below.

Here are some things the authors could do to improve this problem with the manuscript:

1) Change the main table to include columns about replication so that the reader can clearly follow what happened.

We thank the reviewer for this suggestion. We have now reorganized the main results Tables in our manuscript. The new **Table 2** now includes a column summarizing the results of replication in the UK Biobank.

2) Only call the ones that firmly replicate the novel hits. Don't prop up the ones that didn't replicate as possibilities unless you can show that the power was limited (although this seems unlikely since the replication sample size was so large). For anything that you can't get additional data for to replicate, list it as a possibility and not as a novel hit.

We have implemented the reviewer's suggestion and agree it helps greatly to focus the results. The new **Table 2** is focused only on those novel variants that were replicated in the UK Biobank, and also includes a column with these replication results. The new **Table 3** presents information on possible associated variants without replication but with suggestive supporting evidence.

3) One variant is listed as "replicating" in the UKB with an opposite direction of effect. This can make sense as a hypothesis in a GWAS where all the variation was not directly genotyped, but here the original data are from WGS. If the original signal were driven by something else or combination of something elses nearby, then you would be able to see it. If you are going to propose that the UKB data may reflect a replication of some other nearby signal that appears different due to LD and population structure differences, then you need to really investigate what is going on here and make a good case for it. To me, the most likely explanation is that this is a variant that is prone to false associations due to population structure.

We agree that this finding (rs74469188, intronic to *CMIP*) does not constitute replication, particularly in whole genome sequencing. We believe it is important not to exclude this result from our Results or Discussion, particularly given the variant of interest was detected in stratified analysis of African Americans from TOPMed, while the replication was performed in European ancestry samples from the UK Biobank.

We now present a forest plot (**Supplementary Figure 3h**) that shows the directions of effect seen in TOPMed Whites are consistent with those observed in European ancestry-based replication in the UK Biobank, while the directions of effect are largely opposite those in TOPMed African Americans. We present this information in the **Results** (p. 17) and **Discussion** (p. 24) while also noting that it could reflect a false positive association. We choose to highlight this finding to draw attention to the important race/ethnic differences in genetic associations that have been given little attention in many prior GWAS publications.

4) For variants that weren't imputed in the UKB, try the exome data.

We did not perform follow-up using the UK Biobank exome data because the novel single variant associations identified in TOPMed did not include any protein coding variants.

5) For a better study overall, use the UKB exome data in the discovery phase, especially for the rare variant collapsing test. While the current discovery datasets used WGS, the only parts missing from ukb exomes would be any non-coding variants too rare to be imputed, which are for the most part also too rare to have power for discovery in this study regardless.

The UK Biobank Exome discovery effort is a good idea, and warrants a separate project outside the scope of the work reported here. In addition, we emphasize that our project in TOPMed brings unique value that would not be present in a UK Biobank Exome study, including: (a) inclusion of cohorts representing diverse race/ethnicity, allowing for stratified analyses of African Americans, (b) inclusion of a substantial number of COPD cases drawn from COPD-enriched studies, and (c) whole genome sequence data that has allowed us to identify novel rare non-coding

variants demonstrating large effects on lung function. We have now updated the **Discussion** (pp. 23-24, p. 26) to further emphasize these points.

6) The method of using allele counts in the cases to set the cutoff can lead to test statistic inflation. The recommended method to prevent false associations is to determine the allele frequency where one would expect at least x minor alleles in the smaller group (<http://www.nealelab.is/blog/2017/9/11/details-and-considerations-of-the-uk-biobank-gwas>). Also, one place in the paper indicates that your cutoff is 20, another place says 30.

We thank the reviewer for this comment. We have now updated our approach to variant filter for case control analyses so that our TOPMed WGS analyses of COPD case-control status are filtered based on expected heterozygosity count (HC) > 30 in cases (**Methods**, p. 9).

For phenome-wide association analysis (pheWAS) in the UK Biobank, we had previously applied a cutoff of 20 which has been updated to filter on effective HC > 30. (**Methods**, p. 12; **Supplementary Table 17**). For our loss of function analysis, we retained our filter on cumulative minor allele count > 5 in order to examine the potential effects of low frequency variants (**Methods**, p. 11).

Other major comments:

6) SAIGE-LMM wasn't originally made for gene-based collapsing analyses. The way its modeling worked could make collapsed results unreliable. A separate SAIGE-GENE software recently became available, but they explicitly stated in their paper that they didn't have it working for LMM yet. It looks like the most recent version may be able to do both collapsing and LMM though, so please confirm that you are using this latest version and that it is set up to do the test exactly as you have.

We have now updated our gene-based analyses using the new implementation of LMM in SAIGE-GENE, as suggested. The update is noted in the **Methods** (p. 11), with updated results presented in **Supplementary Table 11**.

7) The way the results are presented is difficult to slog through. Please try to summarize more broadly when possible and put the fine details in tables or supplement. The results would also be easier to get through if they more clearly told a story.

We thank the reviewer for this suggestion. We have condensed sections of the **Results** where appropriate, noting that complete results are still available in the Supplementary Tables. In addition, we have created new **Tables 2 and 3** that provide a more comprehensive summary of our WGS discovery and follow-up studies. We believe these new Tables will help to guide readers through the bigger picture aspects of our study design and major results.

Minor comments:

8) Provide information [for] the power in this study.

We have added new **Supplementary Figures 1** and **4** that provide detailed information on the power of our quantitative trait analyses in discovery and replication efforts, respectively.

9) Indicate whether there were any previously known GWAS hits that you missed.

We have added new **Supplementary Tables 5A-5F** in which we present the results from our TOPMed WGS analyses for selected variants identified in prior GWAS studies of lung function and COPD. In addition to demonstrating the extent to which we replicated previously reported variants, the new **Supplementary Table 5** also allows us to comment on consistency of the observed directions of effects across race/ethnic-groups and population-based vs. COPD-enriched studies (**Results**, pp. 14-15).

10) Would like more details on how the ICD9 and 10 codes were combined.

In the **Methods** (p. 12), we now provide a reference to the original paper that combined the ICD9 and ICD10 codes for phenome-wide association (Zhou *et al.*, *Nature Genetics*, 2018. PMID: 30104761), which provides the complete details.

Reviewer #2 (Remarks to the Author):

This is a well written and executed study of genetic variants in pulmonary function and COPD. The authors have assembled an extremely large number of multi-ethnic subjects across multiple studies in which whole genome sequencing has confirmed a number of loci previously identified in GWAS studies, as well as identifying novel loci that they attempt to replicate. Strengths of this effort include the number of African Americans studied, the whole genome approach that allows for more complete coverage of genetic variation, and the replication, co-localization, and eQTL efforts that provide support for their findings. Inclusion of COPD-enriched subjects provides additional power to evaluated genetic contributions to the extreme ends of the phenotype spectrum.

We thank the Reviewer for the careful reading of our manuscript and identification of our study's strengths. We appreciate the constructive comments to which we provide detailed responses below.

There are some additional discussion points for consideration that are briefly mentioned in the discussion but deserve additional attention.

1. The replication datasets do not reflect the populations of the study populations. The numbers of African Americans in the replication dataset is relatively small with no emphasis on COPD-enriched participants as seen in the study population, and the number of Hispanics is large in comparison the study populations. The inability to replicate findings in some instances is likely related to these issues.

We thank the reviewer for this insightful comment. We agree that our replication effort was limited by the mismatch of the discovery and replication cohorts. To address this limitation, we have added a power calculation (**Supplementary Figure 4**) that demonstrates the limited power of our replication effects in non-European ancestry groups (**Results**, p. 20). We have also expanded our **Discussion** on these points (p. 24, p. 26).

2. The phenotypes are not consistent across study populations, i.e., sometimes baseline PFTs are used, sometime most recent, some are pre- and post-bronchodilator use. The potential for misclassification, or at least inconsistencies across study populations should be addressed.

We have updated the phenotype harmonization text in the **Supplementary Text** (p. 2) to clarify that all of the spirometry data used for the current WGS analyses in our manuscript were obtained as pre-bronchodilator measures. We have also provided details in the Supplementary cohort descriptions to clarify that we worked with

investigators from each of the contributing cohorts to determine the most practical strategy to construct a cross-sectional subset of PFTs for each cohort, resulting in use of baseline PFTs for some studies and most recent PFTs for others. We have added a statement to the **Methods** (p. 7) to justify these decisions.

3. *It is unclear why variant associations with smoking initiation and cessation are tested in the replication sample. These are not associations that are reported for the study populations.*

We have updated the **Results** (pp. 19-20), and **Supplementary Text** (p. 6) to clarify the purpose of examining the associations with smoking behavior traits. The main focus on these associations is to determine whether identified variants reflect primary association with lung function, or whether the lung function associations reflect residual confounding with smoking behavior. In the case of variants demonstrating replication of association with lung function, it is particularly of interest to determine whether there is additional evidence of association with smoking behavior.

4. *Some discussion of attributable risk associated with identified variants would provide perspective, as would a discussion of what clinical usefulness these variants might have.*

We thank the reviewer for this pertinent point. We have expanded the **Discussion** in relation to the observed effect sizes for common and rare variants in our study (pp. 23-24, p. 27). As we now note in the text, among our novel WGS variants with replication or other suggestive supporting evidence, we observed modest effect estimates for common variants and comparably large effect estimates for rare variants. These results help to place in context the significance of our reported rare variant associations.

Reviewer #3 (Remarks to the Author):

Zhao X et al. Whole genome sequence analysis of pulmonary function and COPD in 20,000 multi-ethnic participants from the NHLBI trans-omics for precision medicine (TOPMed) program.

This is a well written manuscript where associations were observed between lung function and COPD with 32 loci, among those 22 were novel. Table 2 brings these observations together.

We thank the Reviewer for carefully reading our manuscript and identifying important aspects that should be described more thoroughly. We provide our detailed responses below.

However, how do we bring these observations under a wider biological or even clinical umbrella. ARHGEF17 variants leading to loss of function were associated with increased FEV₁/FVC indicating either normal lung function or pulmonary restriction due to perhaps fibrosis.

We agree with this pertinent point. We have followed up on this finding and note that a burden of *ARHGEF17* pLOF variants was nominally associated with FEV₁ but not FVC (**Results**, p. 19). This result suggests the effect of *ARHGEF17* on increased FEV₁/FVC is more closely related to its effect on increased FEV₁ rather than reduced FVC (pulmonary restriction). We have added to the **Discussion** (pp. 23-24) to comment on the observed directions of effect for *ARHGEF17* and other rare variants from our study.

FTO has mentioned as a new factor without further explanations. This fat mass and obesity-associated protein also known as alpha-ketoglutarate-dependent dioxygenase FTO is an enzyme in humans encoded by the FTO gene located on chromosome 16. As one homolog in the AlkB family proteins, it is the first mRNA demethylase that has been identified.

We have not commented extensively on the *FTO* gene specifically, since we did not have additional evidence by colocalization or otherwise to assign a candidate gene to the variant identified in *FTO*. We now note in the **Discussion** (p. 23) that prior associations with obesity observed at the *FTO* locus, appear to actually affect the *IRX3* and *IRX5* genes (Claussnitzer *et al.*, *NEJM*, 2015. PMID: 26287746).

Regucalcin is another marker that stands out to be important for interstitial lung micro-environment.

We agree with the potential importance of regucalcin. While we have expanded **Discussion** of this gene (p. 25), we are not familiar with the specific references to which the reviewer refers, but would be happy to include them.

Taken all the associations together how much do they explain the occurrence of COPD.

In order to comment more on the population relevance of our findings, we have added **Discussion** on the magnitude of observed effect estimates for both common and rare novel variants from our study (pp. 23-24).

We note that a prior, larger genetic association study for COPD (Sakornsakolpat *et al. Nature Genetics*, 2019. PMID: 30804561) provided an estimate of 7%, which is likely inflated due to use in the discovery population. Thus, our reported associations leave the majority of genetic risk of COPD unexplained, though they do highlight new biologic pathways of interest. Additional consideration of these points is now included in the **Discussion** (p. 27).

To what extent do they point to the most important pathways that may be deregulated by external factors, such as smoke, fine-particle dust. I would clarify and extend this issue in the discussion.

We have annotated the overlap of candidate genes from our study with gene ontology terms implicated by previous GWAS to place our novel genes in context (**Supplementary Table 21**) and have included some **Discussion** of the identified pathways (p. 25). We have further expanded on our treatment of the potential role of smoking for some of the identified variants (**Discussion**, p. 23, p. 25), the role of smoking in the ascertainment of our COPD-enriched samples (**Discussion**, p. 26), and the fact that we did not conduct comprehensive studies examining the role of environmental exposures other than smoking (**Discussion**, p. 27).

REVIEWERS' COMMENTS:

Reviewer #1 (Remarks to the Author):

The authors have responded to my concerns.

Reviewer #2 (Remarks to the Author):

I don't have additional comments. I believe the authors have addressed the previous comments.

Reviewer #3 (Remarks to the Author):

The manuscript has improved a lot in quality with more accurate descriptions especially in the discussion. Although it is not always easy to read, the main messages are clear and truthfull. The authors claimed to have identified 10 known GWAS loci and 20 novel loci. In the discussion still is missing the relation of PIAS1, RGN, and FTO genes with their biology in COPD. PIAS1 gene is involved in pro-inflammatory reactions and development of chronic disease. Regucalcin is a protein that in humans is encoded by the RGN gene and involved in senescence and may be a frailty biomarker. COPD is a disease of the elderly. FTO is indeed involved in energy expenditure and the first RNA demethylase. The question is then how all these altered functions change lung function over time.

However, the authors have incorporated many of the requested items. And I am happy with the end result.

Response to Reviewers

We have revised the manuscript according to the additional feedback from Reviewer #3. We now highlight all changes in the text that reflect these points and suggestions using the tracked changes feature in Word.

Reviewer #3 (Remarks to the Author):

The manuscript has improved a lot in quality with more accurate descriptions especially in the discussion. Although it is not always easy to read, the main messages are clear and truthfull. The authors claimed to have identified 10 known GWAS loci and 20 novel loci.

We thank the Reviewer for noting our changes in response to the initial round of review.

In the discussion still is missing the relation of PIAS1, RGN, and FTO genes with their biology in COPD. PIAS1 gene is involved in pro-inflammatory reactions and development of chronic disease.

We have added to the Discussion relevant references to the role of *PIAS1* in pro-inflammatory processes and chronic disease (p. 18).

Regucalcin is a protein that in humans is encoded by the RGN gene and involved in senescence and may be a frailty biomarker. COPD is a disease of the elderly.

We present in the Discussion a detailed description of regucalcin and multiple references supporting its role in aging (pp. 17-18). While we identified a reference suggesting the use of RGN as a frailty biomarker (Cardoso et al., PMID: 30071357), we have not cited that particular reference in our manuscript, as the suggestion of RGN as a frailty biomarker is based primarily on review of other literature.

FTO is indeed involved in energy expenditure and the first RNA demethylase.

We have added references to the role of *FTO* in energy expenditure and as an RNA demethylase (p. 16).

The question is then how all these altered functions change lung function over time.

In the Discussion (p. 20), we note limitations of our current study, which include our focus on cross-sectional lung function. We anticipate future studies from our group

and others will address identification of genetic factors influencing the change in lung function over time.

However, the authors have incorporated many of the requested items. And I am happy with the end result.

We are glad the Reviewer was happy with our prior revisions. We hope that our additional revisions now address fully all of the Reviewer's points and suggestions.